# EDGE IMPORTANCE INFERENCE TOWARDS NEIGHBORHOOD AWARE GNNS

## ABSTRACT

Comprehensive model tuning and meticulous training for determining proper scope of neighborhood where graph neural networks (GNNs) aggregate information requires high computation overhead and significant human effort. We propose a probabilistic GNN model that captures the expansion of neighborhood scope as a stochastic process and adaptively sample edges to identify critical pathways contributing to generating informative node features. We develop a novel variational inference algorithm to jointly approximate the posterior of the count of neighborhood hops and learn GNN weights while accounting for edge importance. Experiments on multiple benchmarks demonstrate that by adapting the neighborhood scope to a given dataset our model outperforms GNN variants that require grid search or heuristics for neighborhood scope selection.

## 1 INTRODUCTION

Graph neural networks (GNNs) (Kipf & Welling, 2016; Bhagat et al., 2011) gain significant attention in recent years due to their success in various areas, such as social information analysis (Li & Goldwasser, 2019), recommender systems (Ying et al., 2018) and biomedical domain (Kishan et al., 2021; Huang et al., 2020). They are considered essential methods for graph representation learning as GNNs can effectively exploit rich topological information by generating a node's features from its neighborhood.

However, selecting an appropriate scope of the neighborhood where GNNs aggregate information remains an important challenge: a narrow scope that covers a limited range of neighborhoods can hurt the predictive performance, and a broad scope that covers long-range neighborhoods can lead to over-smoothing (Li et al., 2018) and unnecessary complexity. Automatic search algorithms (e.g., grid search) face the same issue since careful design of the search space is a daunting task, and validating large GNN structures incurs high computation and time costs.

Extensive research efforts show that appropriately setting neighborhood scopes for GNNs can be critical to their performance improvement (Abu-El-Haija et al., 2019; Zeng et al., 2021; Veličković et al., 2017). However, prior works mainly focus on designing aggregation schemes via regularization (Srivastava et al., 2014; Rong et al., 2019; Hasanzadeh et al., 2020) or network structures (Xu et al., 2018; Klicpera et al., 2018; Chen et al., 2020). These methods inevitably rely on grid search and heuristics to determine the neighborhood scopes, which leads to heavy tuning and unnecessary model complexity.

In this paper, we propose a probabilistic GNN model inferring the most appropriate neighborhood scope given the graph while aggregating node information. Specifically, we model the expansion of neighborhood scope as a stochastic process by defining a beta process (Broderick et al., 2012) over the count of neighborhood hops to allow it to go to infinity. The beta process induces hopwise activation probabilities and its conjugate Bernoulli process enables us to adaptively sample the edges in the neighborhood. In addition, the importance of the edges is evaluated based on the feature similarity between the adjacent nodes. We can thus identify significant pathways that contribute to the node latent features during training. We propose an efficient variational inference method that jointly approximates the posterior of the neighborhood scopes and learns GNN weights. Our model strikes a balance between the neighborhood scope expansion and the number of activated edges within the neighborhood while providing well-calibrated predictions. It enhances GNN performance across various benchmark datasets, as demonstrated by our experiments.

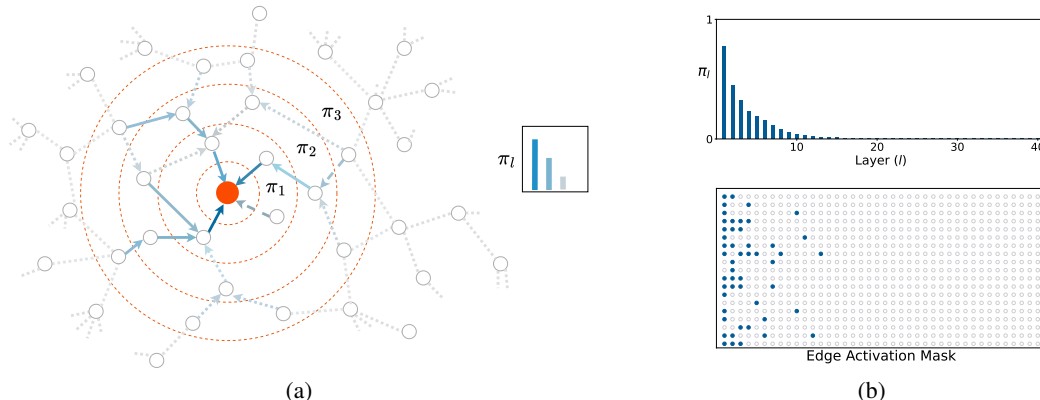

(a)  (b)

Figure 1: (a) Our GNN model jointly infers the proper neighborhood scope and aggregates information through important pathways identified within it. We model the count of neighborhood hops as a beta process to enable it to go infinity. (b) A stick-breaking construction of a beta process and its conjugate Bernoulli process. The sticks located at the top represent random draws from the beta process, serving as layer-wise activation probabilities. Each stick location, denoted by $\delta_{\mathbf{f}_l}$, corresponds to a GNN layer function $\mathbf{f}_l$, with the height indicating its activation probability $\pi_l$. The bottom shows the conjugate Bernoulli process. The binary vectors (column-wise) activate or deactivate edges in each layer by elementwisely multiplying the flattened adjacency matrix.

The contributions of our research are: i) We propose a probabilistic solution for automatically determining the appropriate neighborhood scope for GNNs, which eliminates the need for extensive pre-training and enhances model performance. ii) Our model adaptively samples edges within the neighborhood to identify the important pathways contributing to node encoding. iii) We develop a variational inference to jointly infer the count of neighborhood hops and learn GNN weights. iv) We conduct a comprehensive analysis of our method's neighborhood inference capability and demonstrate its ability to mitigate over-smoothing effectively.

## 2 RELATED WORK

### 2.1 BAYESIAN INFERENCE FOR GNNS

Graph Gaussian Processes (GGPs) (Ng et al., 2018) extends traditional Gaussian Processes (GPs) by incorporating graph topology into the model, enabling it to handle uncertainty in both node features and the graph structure. This approach is designed for situations with limited labeled data, using the graph to propagate information to unlabeled nodes. The Bayesian-GCNN (Pal et al., 2019) framework, as another Bayesian method, interprets the input graph as a single instance drawn from a parametric family of random graphs and estimates the joint posterior distribution of both the graph parameters and the node labels. BBGDC (Beta-Bernoulli Graph DropConnect) (Hasanzadeh et al., 2020) can be viewed as a generalized stochastic Bayesian technique to train GCNs. It enables GCNs to independently drop out edges and convolution channels. However, these methods are not capable of inferring the number of hops automatically during training and require expensive hyper-parameter tuning of network depth.

### 2.2 GNNS WITH MULTI-HOP NEIGHBORHOOD SCOPES

Mixhop (Abu-El-Haija et al., 2019) introduces a novel approach that allows for the mixing of information from different hops in the neighborhood, enhancing the model's ability to learn from local and more distant nodes. It demonstrates how varied neighborhood scopes can be combined to improve representation learning. (Zeng et al., 2021) addresses the limitations of traditional graph neural networks (GNNs) in balancing depth (number of layers) and scope (size of the local neighborhood) by substituting the input graph with a subgraph that preserves essential information. Graph Attention Networks (GATs) (Veličković et al., 2017) introduces the attention mechanism to GCNs, allowing the model to weigh the importance of neighbors during aggregation. By adapting the neighborhood

scope dynamically, GATs improve performance in scenarios where the relevance of neighbors varies significantly.

### 2.3 Edge Importance Evaluation

Dropout (Srivastava et al., 2014) for GNNs is applied to randomly drop node features from the previous hidden layer at each training iteration, based on independent Bernoulli random draws with a constant drop rate. Compared to Dropout, DropEdge (Rong et al., 2019) randomly drops edges from the adjacency matrix instead of node features in each hidden layer during training iterations, based on independent Bernoulli random draws with a constant drop rate. But both methods just randomly choose drop portions, which means they treat each node or edge as equally important ones. Compared with DropEdge, DropEdge++ (Han et al., 2023) introduces a feature-dependent sampler that correlates edge sampling probabilities with the feature similarity of node pairs, determining which edges should be retained or removed.

## 3 Neighborhood-aware GNN

Instead of incurring computational overhead to predetermine the appropriate neighborhood scope for information aggregation, we propose a probabilistic GNN model to automatically infer neighborhood scopes, embodied as network depth, along with identifying important pathways by modeling the count of neighborhood hops as a Beta process over hidden layers while learning GNN weights, as illustrated in Figure 1(a).

### 3.1 Notation

In the following section, $\mathcal{G}(\mathcal{V}, \mathcal{E}, \mathbf{X})$ represents a graph with $N$ nodes/vertices $\mathcal{V}$, edges $\mathcal{E}$, and node features $\mathbf{X}$. $\mathbf{A} \in \mathbb{R}^{N \times N}$ denotes the adjacency matrix of the graph. The adjacency matrix of the graph with added self-connections is denoted by $\tilde{\mathbf{A}} = \mathbf{A} + \mathbf{I}_N$, where $\mathbf{I}_N$ is the identity matrix. Its normalized counterpart is denoted by $\widehat{\mathbf{A}} = \mathbf{D}^{-\frac{1}{2}} \tilde{\mathbf{A}} \mathbf{D}^{\frac{1}{2}}$ where $\mathbf{D}_{ii} = \sum_j \tilde{\mathbf{A}}_{ij}$.

### 3.2 Adaptively Sampling Edges within an Infinite Neighborhood Scope

Let $\mathbf{H}_l$ represent the feature output by the GNN's $l$-th hidden layer for all nodes $\mathcal{V}$. We formulate an infinitely deep GNN with skip-connection as

$$\mathbf{H}_l = \sigma \left( (\widehat{\mathbf{A}} \odot \mathbf{Z}_l) \mathbf{H}_{l-1} \mathbf{W}_l \right) + \mathbf{H}_{l-1}, \quad l \in \{1, \ldots, \infty\} \tag{1}$$

where $\mathbf{W}_l \in \mathbb{R}^{M \times M}$ denotes the weight of layer $l$, with $M$ representing the layer width (i.e., the number of neurons in the layer). Since GNN layer $l$ aggregates information within $l$-th neighborhood hop, we thus adaptively sample edges within $l$-th neighborhood hop by element-wisely multiply (as denoted by $\odot$) the adjacency matrix with a binary matrix $\mathbf{Z}_l$ which is generated from a Bernoulli process, as demonstrated in Figure 1(b).

Given a graph-structured dataset $\mathcal{D} = \{\mathbf{X}, \boldsymbol{y}, \widehat{\mathbf{A}}\}$ with ground-truth labels $\boldsymbol{y}$, the likelihood can be expressed as:

$$p(\mathcal{D}|\mathbf{Z}, \mathbf{W}) = \prod_{n=1}^{N} p(y_n | f_n(\mathbf{X}, \widehat{\mathbf{A}}; \mathbf{Z}, \mathbf{W})) \tag{2}$$

where $y_n$ is the target label and $f_n(\cdot)$ denotes the prediction for the $n^{\text{th}}$ node from the network head, which is softmax for classification. $\mathbf{W} = \{\mathbf{W}_l\}$ denotes the weight tensor, accumulated across the network layers.

### 3.3 Construction of Beta Process Prior

We treat the expansion of neighborhood scope as a stochastic process by modeling the count of neighborhood hops as a Beta process (Paisley et al., 2010; Broderick et al., 2012; KC et al., 2021),

as in Figure 1 (b). A stick-breaking construction of a beta process can be realized as follows:

$$\pi_l = \prod_{j=1}^{l} \nu_j, \qquad \nu_l \sim \text{Beta}(\alpha, \beta) \tag{3}$$

We start by sequentially drawing $\nu_l$ from a beta distribution with hyperparameters $\alpha$ and $\beta$. The cumulative product of $\nu_k$ s until $l$ gives the activation probability for neighborhood hop $l$, which is denoted as $\pi_l$. These probabilities decrease exponentially as $l$ increases, restricting the unbounded growth of the network. We then sample the binary mask for the edges in each neighborhood hop $l$ from a conjugate Bernoulli process $\mathbf{z}_l \sim \text{Ber}(\pi_l)$ which is conjugate to the beta process in Eq. (3). $z_{le} = 1$ activates the edge $e$ in the neighborhood hop $l$ and $z_{le} = 0$ de-activate it. The binary vector $\mathbf{z}_l$ is then reshaped to obtain a full mask matrix $\mathbf{Z}_l$ as in Eq. (1).

Thus, we formulate the prior over the neighborhood hops and the edge sampling matrix $\mathbf{Z}$ as

$$p(\mathbf{Z}, \boldsymbol{\nu}|\alpha, \beta) = p(\boldsymbol{\nu}|\alpha, \beta)p(\mathbf{Z}|\boldsymbol{\nu}) = \prod_{l=1}^{\infty} \text{Beta}(\nu_l|\alpha, \beta) \prod_{e=1}^{|\mathcal{E}|} \text{Ber}(z_{le}|\pi_l) \tag{4}$$

where $\mathbf{Z} = \{\mathbf{Z}_l\}$ and $\boldsymbol{\nu} = \{\nu_l\}$ represent the sets of hop-wise edge masks and activation probabilities, respectively.

### 3.4 MARGINAL LIKELIHOOD FOR NEIGHBORHOOD SCOPE SELECTION

We combine the Beta-Bernoulli process prior in Eq. (4) and the likelihood in Eq. (2), and then marginalize over the edge masks and activation probabilities to obtain the marginal likelihood:

$$p(\mathcal{D}|\mathbf{W}, L, \alpha, \beta) = \int p(\mathcal{D}|\mathbf{Z}, \mathbf{W})p(\mathbf{Z}, \boldsymbol{\nu}|\alpha, \beta)d\mathbf{Z}d\boldsymbol{\nu} \tag{5}$$

### 3.5 VARIATIONAL INFERENCE

Due to the inherent complexity and non-linearity of neural networks, exact marginalization in Eq. (5) over the edge sampling masks is intractable. We propose to approximate it via variational inference.

We adopt the structured variational inference framework (Hoffman & Blei, 2015) to capture the dependency between the activation probabilities and edge sampling masks. We define the variational distribution as

$$q\left(\mathbf{Z}, \boldsymbol{\nu}|\{a_t, b_t\}_{t=1}^{T}\right) = q(\boldsymbol{\nu})q(\mathbf{Z}|\boldsymbol{\nu}) = \prod_{t=1}^{T} \text{Beta}(\nu_t|a_t, b_t) \prod_{m=1}^{N} \prod_{n=1}^{N} \text{ConBer}\left(z_{tmn}|\pi_t; \tau\right) \tag{6}$$

with variational parameters $\{a_t, b_t\}_{t=1}^{T}$. We employ a truncation level $T$ in the variational distribution. Setting $T$ to a sufficiently large number, we can approximate the theoretical assumption of an infinite count of neighborhood hops in the Beta process. We relax the discrete variables by using a concrete Bernoulli distribution $\text{ConBer}(\pi_t; \tau)$ (Maddison et al., 2016; Jang et al., 2016) with temperature parameter $\tau$. This continuous relaxation of the Bernoulli distribution allows back-propagation while sampling the variables.

The evidence lower bound (ELBO) to the marginal likelihood in Eq. (5) is the objective for optimization:

$$\log p(\mathcal{D}|\mathbf{W}, L, \alpha, \beta) \geq \mathbb{E}_{q(\mathbf{Z}, \boldsymbol{\nu})}[\log p(\mathcal{D}|\mathbf{Z}, \mathbf{W})] - D_{\text{KL}}[q(\boldsymbol{\nu})||p(\boldsymbol{\nu})] - D_{\text{KL}}[q(\mathbf{Z}|\boldsymbol{\nu})||p(\mathbf{Z}|\boldsymbol{\nu})] \tag{7}$$

The first term on the RHS is the expectation of the log likelihood with respect to the variational distribution which fits the model to the data. The last two regularization terms are Kullback–Leibler divergence between the model prior and the variational distribution.

### 3.6 EVALUATING THE IMPORTANCE OF EDGES

Sampling the binary edge mask $\mathbf{Z}_l$ results in a random dropping of edges in each layer. However, some edges may be more informative than others for the overall performance of the GNN. For an

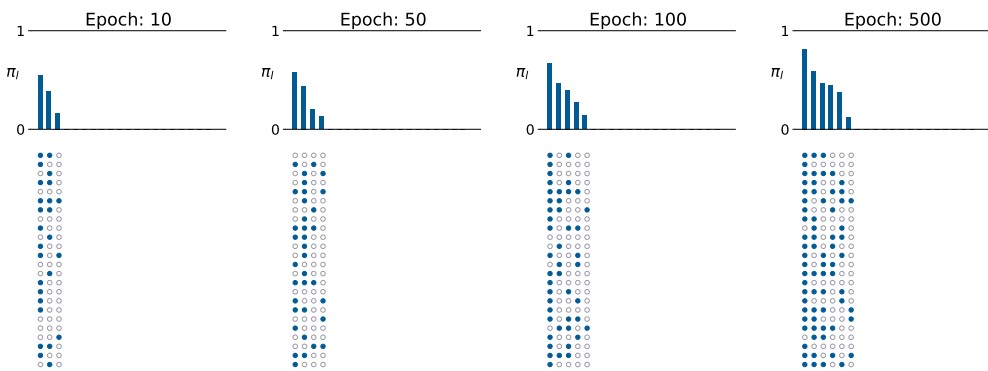

Figure 2: Neighborhood scope evolution and activated edges across different training epochs on Cora dataset during the traing of our method. $\pi_l$ is the hop activation probability (ratio of activated edges) for each neighborhood hop $l$. The neighborhood scope increases as training progresses.

edge $e \in \mathcal{E}$ connecting nodes $n$ and $n'$, we define a kernel function $\kappa(\mathbf{x}_n, \mathbf{x}_{n'})$ to compute the similarity between nodes $n$ and $n'$. Given the hop activation probability $\pi_l$, we weight the edge mask with node similarity as

$$\tilde{z}_{lnn'} \sim \text{Bernoulli}\left(\frac{\pi_l \kappa(\mathbf{x}_n, \mathbf{x}_{n'})}{\sum_{(i,j)\in\mathcal{E}} \kappa(\mathbf{x}_i, \mathbf{x}_j)}\right) \tag{8}$$

This helps preserving the edges whose connecting nodes sharing similar features measured by the kernel. To reduce computational demands during training, we pre-compute the kernel values to avoid recalculating them iteratively.

### 3.7 PREDICTIVE DISTRIBUTION

By applying MAP estimation on the network weights, we obtain the predictive distribution for any node $n$ as follows:

$$p\left(y|\widehat{\mathbf{W}}, \widehat{\boldsymbol{a}}, \widehat{\boldsymbol{b}}\right) = \int p\left(y|f_n\left(\widehat{\mathbf{A}}, \mathbf{x}; \mathbf{Z}, \widehat{\mathbf{W}}\right)\right) q\left(\mathbf{Z}, \boldsymbol{\nu}|\widehat{\boldsymbol{a}}, \widehat{\boldsymbol{b}}\right) d\mathbf{Z}d\boldsymbol{\nu} \tag{9}$$

where, $\widehat{\mathbf{W}}$ is the MAP estimation of network weights and $\widehat{\boldsymbol{a}} = \{\hat{a}_t\}$, and $\widehat{\boldsymbol{b}} = \{\hat{b}_t\}$ denote the optimized varionatal parameters. We perform a Monte Carlo approximation of Eq. (9) by sampling from the variational posterior distribution $q\left(\mathbf{Z}, \boldsymbol{\nu}|\widehat{\boldsymbol{a}}, \widehat{\boldsymbol{b}}\right)$.

## 4 EXPERIMENTS AND DISCUSSION

We analyze the behavior of our proposed probabilistic GNN model on various tasks. First, we illustrate how our model adapts the neighborhood scope during training. Then, we compare our method's performance with GNN variants on the benchmark datasets. These GNN variants rely on grid search to determine the neighborhood hops. Furthermore, we investigate the impact of different kernel functions and evaluate the performance on larger datasets. Along with the ablation study, we assess time complexity, over-smoothing prevention, and uncertainty quantification.

### 4.1 DATASETS

We experiment with three publicly available citation network datasets: Citeseer, Cora, and Pubmed (Sen et al., 2008), as well as two Co-author/Co-purchase network datasets: Co-author CS and Co-author Physics (Shchur et al., 2018), to explore semi-supervised node classification tasks. Additionally, we evaluate the potential of our method on five medium-scale graph datasets: ogb-Arxiv, ogb-Mag (Hu et al., 2020), Flickr (McAuley & Leskovec, 2012), ogb-Proteins and ogb-Products. The details of these datasets are provided in Table 1. All the datasets undergo preprocessing and are

Table 1: The details of the datasets.

| Dataset | #Nodes | #Edges | #Classes | #Features |
|---|---|---|---|---|
| Cora | 2,708 | 5,429 | 7 | 1,433 |
| Citeseer | 3,327 | 4,732 | 6 | 3,703 |
| Pubmed | 19,717 | 44,338 | 3 | 500 |
| CoauthorCS | 18,333 | 163,788 | 15 | 6,805 |
| CoauthorPhysics | 34,493 | 495,924 | 5 | 8,415 |
| Flickr | 89,250 | 899,756 | 7 | 500 |
| ogb-Arxiv | 169,343 | 1,166,243 | 40 | 128 |
| ogb-Mag | 1,939,743 | 21,111,007 | 349 | 128 |
| ogb-Proteins | 132,534 | 39,561,252 | 2 | 8 |
| ogb-Products | 2,449,029 | 61,859,140 | 47 | 100 |

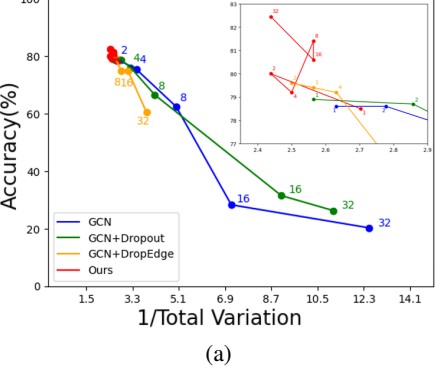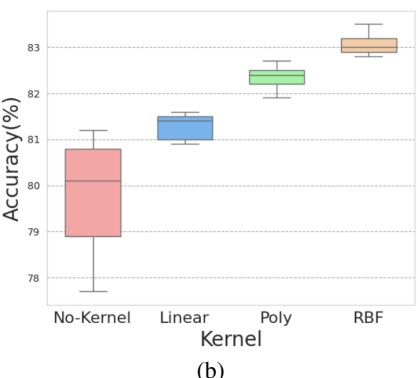

(a)                (b)

Figure 3: (a) Evaluating the effectiveness of our method in preventing over-smoothing. The x-axis represents the inverse of Total Variation (1/TV), as a quantification of over-smoothing, while the y-axis displays the corresponding test accuracy. Each dot on the graph is annotated with the count of neighborhood hops where the models aggregate information. (b) Test accuracies for different kernels on the Cora dataset.

partitioned following (Yang et al., 2016). Within our training set, each class was represented by 20 labeled nodes, totaling 1000 nodes within the test set. Notably, the remaining nodes retained their unlabeled status. Concurrently, for hyperparameter tuning, 500 validation nodes were incorporated, aligning with the approach adopted by (Kipf & Welling, 2016).

## 4.2 EXPERIMENT SETUP

To mitigate potential out-of-memory complications, we adopt a conservative mini-batch size of 10 and a truncation level $K = 2$. The hidden layers of our model incorporate ReLU activation. We use RBF kernel function in our model to evaluate edge importance. We train our models using the Adam optimizer (Kingma & Ba, 2014) with a learning rate of 0.01 and other parameters set as default. We adhere to the original parameters delineated in (Kipf & Welling, 2016), which encompass a layer width consisting of 16 neurons and a dropout probability of 0.5 applied to the hidden layers.

## 4.3 NETWORK STRUCTURE EVOLUTION OVER EPOCHS

We demonstrate how the proposed method inferring neighborhood scope during training on the Cora dataset. The results in Figure 2 show that the expasion of neighborhood hops as training progresses. Specifically, we observe that the activation probabilities per neighborhood hop increase, which, in turn, activates more edges in the graph during later epochs. This graph evolution process continues until it converges to an optimal configuration, after which no further changes in the graph structure are observed.

Table 2: Test accuracy (%) on semi-supervised node classification tasks. The best performances across the benchmark datasets are bolded.

|  | GCN | ResGCN | GCN+DE | JKNet | GCNII | GAT | Ours |
|---|---|---|---|---|---|---|---|
| Cora | 78.7 | 80.9 | 81.2 | 79.7 | **83.8** | 81.7 | 83.2±0.5 |
| Citeseer | 66.2 | 67.3 | 69.3 | 68.9 | 69.8 | 66.0 | **71.5**±0.3 |
| Pubmed | 77.5 | 77.6 | 78.1 | 77.3 | 77.4 | 77.4 | **78.5**±0.2 |
| CoauthorCS | 88.2 | 88.5 | 89.4 | 90.1 | 89.7 | 89.9 | **91.1**±0.2 |
| CoauthorPhysics | 91.4 | 91.7 | 92.2 | 92.1 | 92.7 | 90.8 | **93.1**±0.3 |

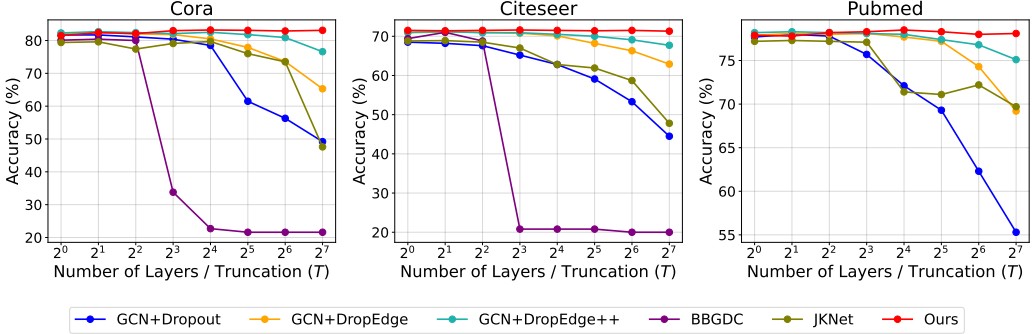

Figure 4: Analysis of the effect of the truncation level $T$ of our method and neighborhood scope of other methods on three citation datasets. The reported performance metric is the classification accuracy (in percentage) on the test sets. The performance of our model (red) is robust. Other methods suffer from over-smoothing.

## 4.4 OVERALL PERFORMANCE COMPARISON

Table 2 presents the semi-supervised learning performance evaluation, highlighting the instances of the best performance. In this evaluation, we compare our method against competing GNN variants on citation and Co-author/Co-purchase network datasets. GCN+DE refers to the vanilla GCN with DropEdge regularization. The competitive performance of GCN+DE shows the advantage of the edge masks, which is consistent with our model's superior performance on all the datasets. Note that there is no statistical significance between our method and GCNII on the Cora dataset.

## 4.5 THE MECHANISMS MITIGATING OVER-SMOOTHING

We examine the total variation (TV) of the outputs from our model's hidden layers throughout the training process. TV quantifies the smoothness of a signal distributed across the nodes of a graph (Chen et al., 2015). Specifically, given a graph with adjacency matrix $\mathbf{A}$ and a signal $\mathbf{H}$ defined across its nodes, the TV is defined as $\text{TV}(\mathbf{H}) = \|\mathbf{H} - 1/|\lambda_{max}|\mathbf{A}\mathbf{H}\|_2^2$ where, $\lambda_{max}$ denotes the eigenvalue of the adjacency matrix $\mathbf{A}$ with the largest magnitude. A lower TV indicates that the signal on adjacent nodes is more consistent across orders, serving as an indicator of the presence of the over-smoothing problem.

Figure 3(a) shows the effectiveness of our method in preventing over-smoothing compared to other regularization techniques. In this experiment, we compare our method against vanilla GCN, GCN with dropout regularization, and GCN with DropEdge regularization. The results show that vanilla GCN and GCN with dropout suffer from a more pronounced over-smoothing issue. The total variation decreases as these models aggregate information from long-range neighborhoods and lead to a rapid decline in test accuracy. On the other hand, GCN with DropEdge partially alleviates this oversmoothing problem as the total variation is less impacted compared to the previous two models. In contrast, our method demonstrates superior effectiveness and robustness, particularly for large neighborhood scopes.

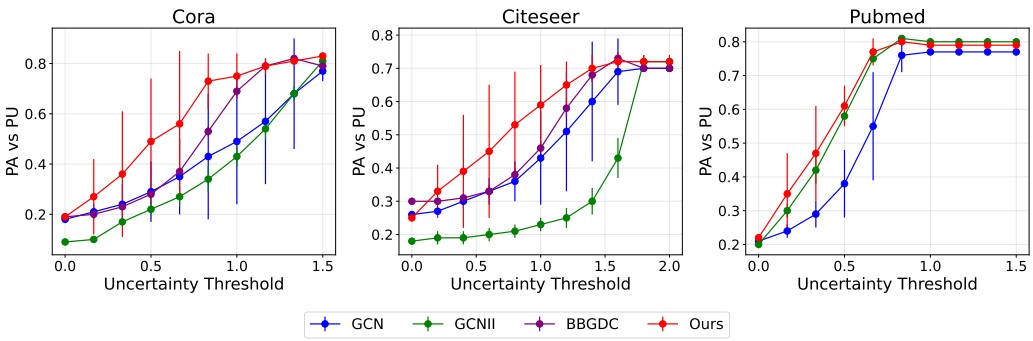

Figure 5: Evaluation of uncertainty estimation on the three citation datasets. The x-axis represents the uncertainty threshold, which discerns the demarcation point between certain and uncertain predictions. Plotted on the y-axis is the PAvsPU metric (Mukhoti & Gal, 2019), which reveals the increasing trend of correctly estimated uncertainty as thresholds rise.

## 4.6 KERNEL FUNCTION EVALUATION

We evaluate edge importance using three kernels: linear, polynomial, and Radial basis function (RBF) kernels, which are represented by $\kappa_{\text{linear}}(\mathbf{x}_n, \mathbf{x}_{n'}) = \mathbf{x}_n^T \mathbf{x}_{n'}$; $\kappa_{\text{poly}}(\mathbf{x}_n, \mathbf{x}_{n'}; n) = (\mathbf{x}_n^T \mathbf{x}_{n'})^n$; $\kappa_{\text{RBF}}(\mathbf{x}_n, \mathbf{x}_{n'}; \gamma) = \exp(-\gamma \|\mathbf{x}_n - \mathbf{x}_{n'}\|^2)$, respectively. Figure 3(b) shows the results of using different kernels. The model without a kernel function is the one we disregard edge importance. This leads to significant fluctuations in performance as observed by the large variation for the test accuracy. This instability arises because, during training, important edges are randomly dropped, resulting in a loss of crucial information. This issue becomes particularly problematic when the dataset is small or the edges are sparsely distributed. Applying kernels significantly stabilizes performance as observed by reduced variations in the test accuracies. For the polynomial and RBF kernels, we learn the parameters degree $n$ and the parameter $\gamma$, respectively. The RBF kernel function outperforms the other two kernel functions.

## 4.7 PERFORMANCE ON LARGER DATASETS

We also test our method on five medium-scale graph datasets: Flickr, ogb-Arxiv, ogb-Mag, ogb-Proteins and ogb-Products to assess the scalability of our method. Table 3 shows the performance compared to different baseline methods. Our model achieves the best performance on these datasets, as we retain critical information while dropping out redundant edges. Note that rather than relying on any search algorithms to determine the neighborhood scopes, our model automatically balances the neighborhood scope and the activated pathways within it while learning GNN weights on these larger dataset.

Table 3: Test accuracy (%) comparisons with larger datasets on semi-supervised node classification task. The results are the best performance achieved by the baseline methods.

|  | GCN | ResGCN | JKNet | GCNII | GAT | Ours |
|---|---|---|---|---|---|---|
| Flickr | 51.4 | 51.9 | 51.6 | 46.1 | 52.2 | **53.5 ± 1.3** |
| ogb-Arxiv | 72.1 | 72.3 | 72.2 | 72.7 | 73.6 | **75.2 ± 0.4** |
| ogb-Mag | 37.3 | 37.9 | 38.4 | 42.3 | 43.7 | **44.3 ± 1.7** |
| ogb-Proteins | 72.5 | 73.4 | 69.5 | 74.1 | **85.0** | 83.6 ± 0.3 |
| ogb-Products | 82.3 | 82.5 | 82.9 | 83.7 | 81.7 | **83.8 ± 0.4** |

## 4.8 ABLATION STUDY

To evaluate the effectiveness of each module in our model, we present the results of an ablation study in Table 4. By comparing the vanilla GCN model (w/o kernel, beta process, and skip-connection) and our model, we assess the contribution of individual components, including skip connection, the beta process, and the kernel function. The results indicate that the beta process significantly

enhances performance, particularly on the Citeseer dataset, where using the beta process alone yields improved results. However, the performance is not stable without the kernel function, indicated by larger standard deviation. Incorporating the kernel function not only stabilizes the outcomes but also statistically improves performance.

Table 4: Ablation study of different modules' effectiveness in our model. The best performance is bolded.

|  | Cora | Citeseer | Pubmed | ogb-Arxiv | ogb-Mag |
|---|---|---|---|---|---|
| Ours | $\mathbf{83.2 \pm 0.5}$ | $71.5 \pm 0.3$ | $\mathbf{78.5 \pm 0.2}$ | $\mathbf{75.2 \pm 0.4}$ | $\mathbf{44.3 \pm 1.7}$ |
| w/o kernel | $82.2 \pm 1.2$ | $\mathbf{71.7 \pm 1.1}$ | $77.9 \pm 0.6$ | $74.3 \pm 1.1$ | $42.7 \pm 3.1$ |
| w/o beta process | $79.4 \pm 0.3$ | $67.8 \pm 0.2$ | $77.9 \pm 0.2$ | $72.3 \pm 0.2$ | $43.1 \pm 0.7$ |
| w/o skip-connection | $81.2 \pm 0.4$ | $69.8 \pm 0.1$ | $77.6 \pm 0.3$ | $71.5 \pm 0.6$ | $42.9 \pm 0.4$ |
| w/o kernel, beta process and skip-connection | $78.7 \pm 0.2$ | $66.2 \pm 0.3$ | $77.5 \pm 0.2$ | $70.2 \pm 1.3$ | $37.2 \pm 0.6$ |

## 4.9 TIME COMPLEXITY ANALYSIS

For training a GCN structure with depth $L$ and width $M$, the time complexity of our method is $O(NBLM^2)$ where $N$ represents number of training nodes and $B$ represents number of epochs. Let $T$ be the time cost of a single forward pass, with $S$ samples our method is linearly scalable as $ST$.

We conduct a comparison of the training time costs using the same mini-batch size. Table 5 shows the results for $S = 1$. More experiment results with different settings of sample $S$ can be seen in the Appendix. Although our method requires more time during training, it automatically infers the optimal neighborhood scope, eliminating the overhead with automatic search algorithms. The time and space consumption during training is still comparable to the baseline methods, demonstrating the efficiency of our approach.

Table 5: Semi-supervised node classification training time comparison between vanilla GCN and ours. The training time unit is in seconds (s) and the space unit is in Megabyte (MB).

| Methods | Cora | | Citeseer | | Pubmed | | ogb-Arxiv | | ogb-Mag | |
|---|---|---|---|---|---|---|---|---|---|---|
| | Time | Space | Time | Space | Time | Space | Time | Space | Time | Space |
| GCN | 62.53 | 39 | 77.37 | 164 | 96.54 | 129 | 643.72 | 2421 | 2765.45 | 4805 |
| GCNII | 60.14 | 42 | 79.22 | 175 | 95.76 | 133 | 614.37 | 2525 | 2840.32 | 4953 |
| JKNet | 61.27 | 41 | 80.15 | 184 | 100.33 | 136 | 661.44 | 2606 | 2911.59 | 4904 |
| Ours | 67.38 | 57 | 88.51 | 194 | 102.36 | 166 | 677.58 | 2788 | 3033.75 | 5277 |

## 4.10 OVER-SMOOTHING PREVENTION

To illustrate the effectiveness of our approach in alleviating the over-smoothing problem, we showcase the effectiveness of our method in counteracting the over-smoothing issue by comparing it with other techniques. Figure 4 illustrates the changes in prediction accuracy as we expand neighborhood scope to aggregate more information in our method. In addition to comparing with benchmark models, we also evaluate various regularization techniques, including Dropout, DropEdge, and DropEdge++. Our method, which decouples the truncation level from the neighborhood order of data aggregation, is presented to be the most robust technique, maintaining consistent performance as the truncation level increases. This demonstrates its superior ability to mitigate over-smoothing compared to other techniques.

## 4.11 UNCERTAINTY QUANTIFICATION

We compare our method with vanilla GCN, GCNII, and BBGDC (Hasanzadeh et al., 2020) to assess uncertainty quantification. First, we evaluate using the PAvsPU metric (Mukhoti & Gal, 2019). The horizontal axis represents the uncertainty threshold, which delineates predictions deemed certain or uncertain. Predictions falling below the threshold are deemed certain, while those surpassing it are

considered uncertain. The PAvsPU metric captures the proportion of correctly estimated uncertainties relative to all model predictions made on the test dataset. Figure 5 shows the results of this metric across different thresholds, showing our method's capability of better uncertainty estimation.

Table 6: Evaluating the uncertainty estimation of models with ECE metric.

|  | Cora | Citeseer | Pubmed | ogb-Arxiv | ogb-Mag |
|---|---|---|---|---|---|
| GCN | $23.51 \pm 1.89$ | $21.80 \pm 1.21$ | $10.62 \pm 1.28$ | $8.43 \pm 1.22$ | $6.56 \pm 0.97$ |
| GCNII | $26.14 \pm 0.16$ | $28.96 \pm 0.48$ | $15.24 \pm 0.06$ | $10.04 \pm 0.08$ | $6.92 \pm 0.04$ |
| BBGDC | $14.57 \pm 0.33$ | $20.58 \pm 0.12$ | OOM | OOM | OOM |
| Ours | $\mathbf{6.49 \pm 1.64}$ | $\mathbf{14.62 \pm 1.39}$ | $\mathbf{4.97 \pm 0.92}$ | $\mathbf{5.02 \pm 1.03}$ | $\mathbf{3.53 \pm 0.88}$ |

Besides PAvsPU, we also employ the Expected Calibration Error (ECE) (Naeini et al., 2015) to evaluate our model. A lower ECE indicates better calibration, meaning that the predicted probabilities more accurately reflect the true likelihood of outcomes. Table 6 presents the results measured by the ECE metric. Our method consistently demonstrates better performance in both metrics, highlighting its effectiveness in assessing its own confidence or uncertainty in predictions, thereby providing reliable uncertainty estimates.

## 5 CONCLUSION

We introduce a neighborhood-aware GNN model with adaptive edge sampling. Our method leverages the power of the beta process that enables us to determine the appropriate neighborhood scope based on the given dataset. Additionally, we also utilize kernel functions to discover important pathways within the neighborhood. This approach eliminates the computational overhead caused by grid search. The experimental results showcase the effectiveness of our method in enhancing the performance across various datasets.

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

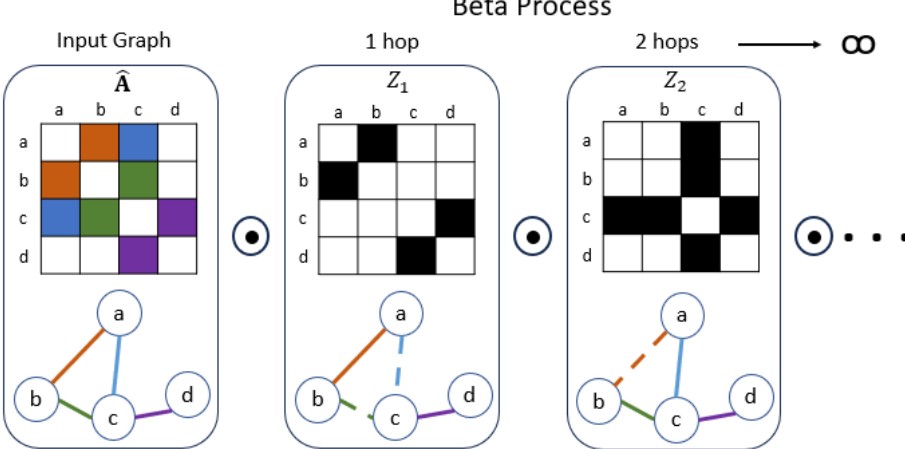

Figure 6: The architecture of our proposed model with a potentially infinite number of neighborhood scope. We use a beta process to enable the neighborhood scope to go infinity. In the $l$-th neighborhood hop, we generate a binary matrix $\mathbf{Z}_l$ as a mask by using the conjugate Bernoulli process to drop edges from the adjacency matrix.

## A    DETAILS ON NEIGHBORHOOD SCOPE INFERENCE OF GNN

The architecture of our proposed model with a potentially infinite number of neighborhood hops is shown in Figure 6. To compute the expectation in Eq. (6) samples the mask, we employ Monte Carlo estimation with $S$ samples of the edge sampling matrix $\mathbf{Z}^{(s)}$, which represents the sets of hop-wise edge masks along with the hop-wise activation probabilities. We utilize the Beta and the Bernoulli processes, where the activation probability decreases exponentially as the value of $l$ increases. For neighborhood hop with large enough $l$ values, the activation probability becomes small, resulting in no edges being activated in that neighborhood hop. The count of neighborhood hops with activated edges is then determined as:

$$l^c = \max_l \left\{ l \bigg| \sum_{m=1}^{|\mathcal{E}|} z_{lm} > 1 \right\} \tag{10}$$

where $\sum_{m=1}^{|\mathcal{E}|} z_{lm}$ represents total activation of edges in neighborhood hop $l$. We can compute the expectation of log-likelihood based on $S$ samples of GNN structure:

$$\mathbb{E}_{q(\mathbf{Z},\nu)}[\log p(\mathcal{D}|\mathbf{Z},\mathbf{W})] \approx \frac{1}{S} \sum_{s=1}^{S} [\log p(\mathcal{D}|\mathbf{Z}^{(s)},\mathbf{W})], \tag{11}$$

where $\mathbf{Z}^{(s)}$ are sampled from the Bernoulli process. The variational parameters $\{a_t, b_t\}_{t=1}^{l^c}$ and $\{\mathbf{W}\}_{t=1}^{l^c}$ are learned through an end-to-end optimization of the ELBO as depicted in Eq. (7).

## B    MORE EXPERIMENTS

### B.1    TIME COMPLEXITY ANALYSIS WITH DIFFERENT SAMPLE COUNTS

Table 7 presents additional results from the time consumption experiments. Compared to the vanilla GCN method, our approach shows a linear increase in time consumption, which corroborates the earlier time complexity analysis. Specifically, with $S$ samples, our method scales linearly as $ST$, where $T$ represents the time for single pass.

### B.2    DIRICHLET ENERGY ANALYSIS

Dirichlet energy (Zhou et al., 2021) of node embeddings is defined to measure their smoothness within a graph [1]. This formulation captures the total variation of the embeddings across con-

Table 7: Semi-supervised node classification training time comparison with different sample $S$ settings. The training time unit is in seconds(s)

|  | Cora | Citeseer | Pubmed |
|---|---|---|---|
| GCN | 62.53 | 77.37 | 96.54 |
| GCNII | 60.14 | 79.22 | 95.76 |
| Ours $S = 1$ | 67.38 | 88.51 | 102.36 |
| Ours $S = 3$ | 205.45 | 263.13 | 298.96 |
| Ours $S = 5$ | 340.72 | 438.17 | 506.45 |

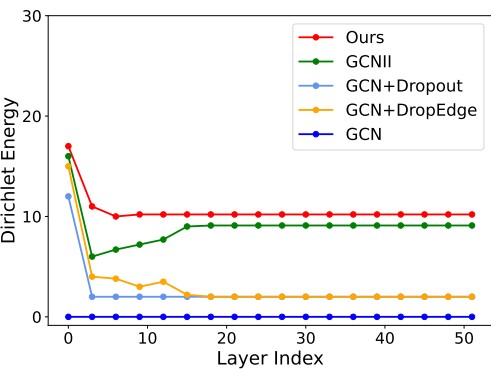

Figure 7: Dirichlet Energy variation with layers in Cora dataset.

nected nodes, serving as an indicator of how much the embeddings change between adjacent nodes. In Graph Neural Networks (GNNs), monitoring the Dirichlet energy across layers helps in understanding and mitigating the over-smoothing phenomenon, where node embeddings become overly similar, leading to a loss of discriminative power.

Figure 7 shows the variation of Dirichlet energy with increasing layers in Cora dataset. The higher value indicates that the node embeddings are over-separating even for those nodes with the same value, while the lower value shows the presence of over-smoothing problem. Our method shows higher Dirichlet energy than other methods as we increase the truncation level in our method and layers in other methods. The results is consistency with the ones we showed using total variation.

### B.3 Neighborhood Scope Inference Analysis on Synthetic Dataset

In this section, we present a detailed analysis of neighborhood determination using two synthetic datasets: BA-Shapes and Tree-Cycles, which are commonly utilized in GNN explanation experiments (Ying et al., 2019). These datasets are selected because they provide ground-truth explanations in the form of motifs, which can be viewed as subgraphs that contribute to predictions in various tasks.

Figure 8 illustrates the results for both synthetic datasets. The inferred neighborhood scope identifies the most suitable range for extracting meaningful information during training for each dataset. For example, in the BA-Shapes dataset, the motif indicates that up to second-order neighborhoods are sufficient to provide adequate information for node classification tasks. Similarly, in the Tree-Cycles dataset, the motif, which forms a ring structure, suggests that up to third-order neighborhoods are necessary to capture the required information.

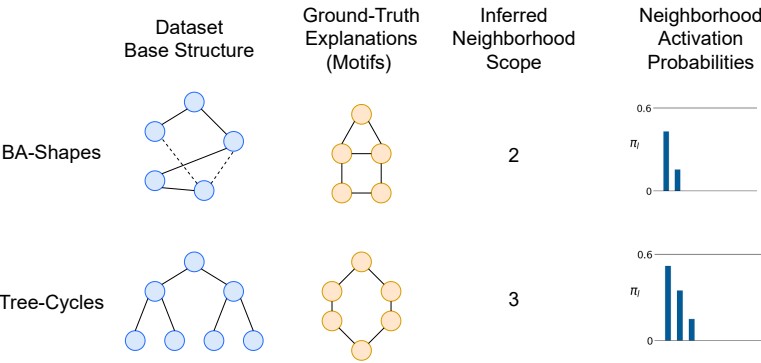

Figure 8: Neighborhood scope inference results and inferred neighborhood activation probabilities for BA-Shapes and Tree-Cycles datasets.

