# OpenReview forum: "Edge Importance Inference Towards Neighborhood Aware GNNs"
_ICLR.cc/2025/Conference — Submitted to ICLR 2025_

### Official Review · Reviewer_MmuU · 2024-10-30

**Soundness:** 2
**Presentation:** 2
**Contribution:** 2
**Rating:** 5
**Confidence:** 4

**Summary:**

In this paper, the authors propose a neighborhood-aware GNN model with adaptive edge sampling. Specifically, they use a variational inference to jointly infer the count of neighborhood hops and learn GNN weights. Experiments conducted on the node classification task showcase the effectiveness of their method.

**Strengths:**

1. Multi-dimensional model evaluation (effect, time, uncertainty, over-smoothing);
2. The principled approach;
3. The paper is well-organized.

**Weaknesses:**

1. The experimental comparison is insufficient. First, there is no introduction to the comparison method, and baselines are not new enough. Consider adding a recent method comparison. In addition, as an improvement to the basic GNN model, it is not enough to experiment only on node classification.
2. Does this method work effectively for heterophilic graphs?
3. Consider adding further theoretical support or more visualization results of neighbor modeling

**Questions:**

Please see the weaknesses.

---

> ### Author Response · Authors · 2024-11-25
>
> We thank Reviewer MmuU for the constructive feedback.
>
> **"No Introduction to the Comparison Method":** Most of the baseline methods including the state-of-the-art are discussed in Section 1 Introduction and Section 2 Related Works. We will add the discussion about the missing ones: GCNII and JKNet in the camera-ready version.
>
> **"Experiment with a recent method":** We have experimented with a most recent method named PSNR-GNN [1]. The resulted reported in table below demonstrate that our method achieves better predictive accuracy across the benchmark datasets.
>
> | Dataset   | PSNR-GNN    | Ours        |
> |-----------|-------------|-------------|
> | Cora      | 82.8±3.6    | **83.2±0.5**    |
> | Citeseer  | 71.2±0.6    | **71.5±0.3**   |
> | Pubmed    | 77.9±1.5    | **78.5±0.2**   |
>
> We will add the additional results and discussions in the camera-ready version.
>
> **"Experiments other than node classification":** Although the main focus of our work is on node classification, intuitively we can extend it to solve other graph-related problems, such as graph classification. Since our method adaptively sample edge during learning which may reduce the number of link examples, its extension to link prediction might be a little problematic. This is a part of our future work.
>
> **Heterophilic graph experiments:** As the reviewer suggested, we conduct additional experiments on heterophilic graphs. We hypothesize that one main reason why our method doesn’t perform ideally on these datasets is because of the random edge sampling during training. The dropped edges may lead to information loss about the connections between nodes with different labels. This information loss can cause significant impact due to the small sizes of the heterophilic datasets. We have the results in the table below for reference.
>
> | Dataset    |   GCN   | GAT+JK  |   Ours   |
> |------------|----------|----------|----------|
> | Cornell    |   82.4   |   66.5   |   62.4   |
> | Wisconsin  |   75.5   |   69.5   |   60.1   |
> | Texas      |   83.1   |   75.4   |   62.7   |
> | Chameleon  |   64.1   |   68.1   |   55.2   |
>
>
> **More visualization Results on Neighborhood modeling:** We have added an analysis on neighborhood scope inference in [this link](https://anonymous.4open.science/r/iclr2025-D60E/neighborhood%20modeling.pdf) and also in Appendix B.3 in the updated version. We use two synthetic datasets, BA-Shapes and Tree-Cycles, used in GNN explanation studies[2]. Analyzing their ground-truth explanations in the form of motifs, we conclude that inferred neighborhood scope highlights the optimal range for extracting meaningful information during training, with second-order neighborhoods being sufficient for BA-Shapes and third-order neighborhoods required for the ring-like motifs in Tree-Cycles.
>
>
> [1] Jingbo Zhou, Yixuan Du, Ruqiong Zhang, Jun Xia, Zhizhi Yu, Zelin Zang, Di Jin, Carl Yang, Rui Zhang, & Stan Z. Li (2024). Deep Graph Neural Networks via Posteriori-Sampling-based Node-Adaptative Residual Module. In The Thirty-eighth Annual Conference on Neural Information Processing Systems.
>
> [2]​​ Ying, Z., Bourgeois, D., You, J., Zitnik, M., & Leskovec, J. (2019). Gnnexplainer: Generating explanations for graph neural networks. Advances in neural information processing systems, 32.

---

> ### Author Response · Authors · 2024-12-02
> **Gentle Remainder**
>
> We thank the reviewer for the constructive feedback and would like to respectfully remind the reviewer that the deadline for the discussion period is approaching. if the reviewer has any other questions, we would be glad to address them. Thank you again for your time and thoughtful review!

---

> ### Author Response · Authors · 2024-12-03
>
> **"Experiments other than node classification"**: As Reviewer MmuU suggested, we extend our method for a graph classification task by incorporating a global pooling layer after the GNN layers to aggregate node features into a single graph representation for classification. We evaluate our method on Proteins [1] dataset, and the results [2] are shown in the table below.
>
> | Method| Proteins    |
> |-----------|-------------|
> | GAT       | 70.9±2.7    |
> | GraphSAGE | 73.0±4.5    |
> | PathNN-$\mathcal{P}$ | **75.2±3.9**    |
> | Ours      | 71.6±2.3    |
>
> Our proposed work primarily focuses on node classification. While our method's performance on graph classification is not particularly outstanding, the experiments demonstrate that our approach can be extended to address other graph-related problems.
>
> [1] Morris, C., Kriege, N. M., Bause, F., Kersting, K., Mutzel, P., & Neumann, M. (2020). Tudataset: A collection of benchmark datasets for learning with graphs. arXiv preprint arXiv:2007.08663.
> [2] Michel, G., Nikolentzos, G., Lutzeyer, J. F., & Vazirgiannis, M. (2023, July). Path neural networks: Expressive and accurate graph neural networks. In International Conference on Machine Learning (pp. 24737-24755). PMLR.

---

> ### Author Response · Authors · 2024-12-04
>
> We have made every effort to address all the concerns and questions raised. As the discussion period is nearing its end, we kindly request a consideration from Reviewer MmuU for an updated score. If there are any further questions or matters to discuss, please don’t hesitate to let us know.

---

### Official Review · Reviewer_qxoW · 2024-11-04

**Soundness:** 2
**Presentation:** 3
**Contribution:** 2
**Rating:** 5
**Confidence:** 3

**Summary:**

The paper introduces a framework that optimizes neighborhood scope selection in GNNs. This method adaptively samples edges within the neighborhood, allowing for the identification of critical pathways that contribute to effective node encoding. It is claimed to reduce computational overhead while also enhancing the GNN’s ability to capture relevant structural information.

**Strengths:**

The idea is novel, with a clear and reasonable motivation. The experiment structure is appropriate, and the results are solid.

**Weaknesses:**

The paper has several presentation issues, including inconsistent citation formatting (e.g., not using \citep) and a lack of uniformity in text font and size across figures. Additionally, the methods section is difficult to follow, making it challenging to fully understand the motivation and logic behind the proposed approach. Consistency among baselines is also needed across relevant methods; for instance, the time complexity table should include all over-smoothing methods to provide a complete comparison.

**Questions:**

- See weaknesses.
- It’s unclear why the method requires both $\mathbf{Z}$ and $\mathbf{\nu}$ to capture the importance of an edge.

---

> ### Author Response · Authors · 2024-11-25
>
> We thank Reviewer qxoW for the constructive feedback.
>
> **"Citation Formatting and Inconsistent Figure Text Size Issues":** We fix the problems, pointed out by the reviewer, in the revised version. We will include the changes in the camera-ready version.
>
> **"Consistency across Baselines":** We conduct additional experiments to include the larger datasets, such as ogbn-proteins and ogbn-products in tables 3, 4, 5 and 6, for comprehensive comparison. These additions strengthen the validation of our approach and demonstrate its effectiveness across a broader range of dataset scales. And, for reference, we have included a portion of these experiments on these larger datasets in the table below.
>
> | Dataset/Metric         | GCN   | ResGCN | GCNII | GAT   | JKNet | Ours            |
> |-------------------------|-------|--------|-------|-------|-------|-----------------|
> | OGB-proteins (AUC-ROC)  | 72.5  | 73.4   | 74.1  | **85.0**  | 69.5  | 83.6 ± 0.3      |
> | OGB-products (Accuracy) | 82.3  | 82.5   | 83.7  | 81.7  | 82.9  | **83.8 ± 0.4**      |
>
> We have further added ablation study with larger datasets, which is shown below.
> | Methods                                     | ogb-Arxiv          | ogb-Mag           |
> |---------------------------------------------|--------------------|-------------------|
> | Ours                                        | **75.2 ± 0.4**     | **44.3 ± 1.7**    |
> | w/o kernel                                  | 74.3 ± 1.1         | 42.7 ± 3.1        |
> | w/o beta process                            | 72.3 ± 0.2         | 43.1 ± 0.7        |
> | w/o skip-connection                         | 71.5 ± 0.6         | 42.9 ± 0.4        |
> | w/o kernel, beta process, and skip-connection | 70.2 ± 1.3         | 37.2 ± 0.6        |
>
>
>
>
> Also, we have added time complexity for larger dataset (ogb-Mag) in table 5, which is shown  below (italic typeface used to represent added results).
> | Methods    | Time (Cora) | Space (Cora) | Time (Citeseer) | Space (Citeseer) | Time (Pubmed) | Space (Pubmed) | Time (ogb-Arxiv) | Space (ogb-Arxiv) | Time (ogb-Mag) | Space (ogb-Mag) |
> |------------|-------------|--------------|------------------|------------------|---------------|----------------|------------------|------------------|----------------|----------------|
> | GCN        | 62.53       | 39           | 77.37           | 164              | 96.54         | 129            | 643.72           | 2421             | *2765.45*      | *4805*         |
> | GCNII      | 60.14       | 42           | 79.22           | 175              | 95.76         | 133            | 614.37           | 2525             | *2840.32*      | *4953*         |
> | *JKNet*    | *61.27*     | *41*         | *80.15*         | *184*            | *100.33*      | *136*          | *661.44*         | *2606*           | *2911.59*      | *4904*         |
> | Ours       | 67.38       | 57           | 88.51           | 194              | 102.36        | 166            | 677.58           | 2788             | *3033.75*      | *5277*         |
>
>
> **"Capturing Edge Importance using $\mathbf{Z}$ and $\boldsymbol{\nu}$":** $\mathbf{Z}$ and $\boldsymbol{\nu}$ adaptively sample important edges within the neighborhood. Meanwhile, we jointly learn the kernel functions to evaluate the importance level of these edges, as described in section 3.6. We will clarify these in the camera-ready version.

---

> > ### Author Response · Authors · 2024-12-02
> > **Gentle Remainder**
> >
> > We thank the reviewer for the constructive feedback and would like to respectfully remind the reviewer that the rebuttal deadline is approaching. if the reviewer has any other questions, we would be glad to address them. Thank you again for your time and thoughtful review!

---

> > ### Comment · Reviewer_qxoW · 2024-12-02
> >
> > Thank you for your detailed response, which addressed all my concerns and included extended experiments. I have no further questions at this time. I will increase my score to 5.

---

> ### Author Response · Authors · 2024-12-03
>
> We are quite grateful to Reviewer qxoW for raising the score. Since you mentioned that we have addressed all your concerns and questions, we wonder if it would be possible for you to kindly increase the score further.
>
> If there are any further questions or matters to discuss, please don’t hesitate to let us know.

---

### Official Review · Reviewer_g91e · 2024-11-04

**Soundness:** 3
**Presentation:** 2
**Contribution:** 3
**Rating:** 5
**Confidence:** 4

**Summary:**

This work considers the problem of determining the neighborhood scope of a problem. In other words, determining the contribution of nodes at different distances from the target. To tackle the inefficiency of the standard train-then-validate approach, a technique called *Bayesian Neighborhood Aggregation* (BNA) is presented, which uses a beta process prior on the contributions of nodes at different distances, and a conjugate Bernoulli distribution over the features dimensions used for message passing. The data likelihood is jointly optimized over the (mean-field) variational distribution and the model weights.

**Strengths:**

1. The idea of using a Bayesian model for inferring the neighborhood scope is novel.

2. The empirical analysis is quite thorough, with an evaluation of model performance, over-smoothing analysis, effect of kernel choice, performance on small and large datasets, and time and memory costs.

3. The model performs well, or at least competitively, in comparison to other baselines, in various aspects like the ones mentioned above.

**Weaknesses:**

1. The literature review in Section 2 looks weak because the description of the works is more about the proposed algorithms, but not their relevance to the current work. As in, the descriptions look like "ABC work does XYZ", but their strengths, weaknesses and/or relevance to BNA are not discussed. Another point is that despite the extensive literature review, in Section 4, no comparison is made with the Bayesian GNN methods, only with the architectural changes.

2. A modelling intuition, including strengths and weaknesses, is not presented for the choice of the beta process prior. All I understand is that a low $\nu_j$ reduces the weight of nodes at $j$-hops and further away, so the model assumes that the message importance decays monotonically as distance between nodes increases.

3. I am getting the impression that $\pi_l$ is the probability of receiving messages over each feature dimension from nodes **up to** $l$-hops away, and not exactly $l$-hops away, since the augmented adjacency matrix (with self-loops) is used for message passing at each step.

4. While the performance on benchmarks is competitive with the baselines, I would have been more interested in seeing conclusive evidence for neighborhood scope determination, perhaps using a synthetic dataset, where scope of the underlying ground-truth is known. All the other benefits are well and good, but I don't see compelling empirical evidence for automatic determination of neighborhood scope.

**Questions:**

1. What is the advantage of using a conjugate Bernoulli-process for the feature masks? Analytical tractability is anyway not available, and the choice of conjugate prior does not exactly help with the variational learning, does it? Can you also use other masks on the feature matrix, maybe like the one in DropMessage, where all elements in the feature matrix are masked iid?

2. The degree matrix defined near line 132 should be in bold.

3. Kindly add the dimensions of the vectors and the matrices, as well as the spaces they belong to. For example, $\mathbf{H}_l \in \mathbb{R}^{N\times O}$.

4. How do the results in Figure 4 change, when the residual connections are not used? I am hoping to isolate the effect of BNA.

5. What does "Ours" correspond to in Table 2? Which kernel is being used, which architecture?

6. Possible typos (correct me if I am wrong; in that case, I may have read the related parts wrongly):
  - I think you are using \cite{...} everywhere, even where \citep{...} is more appropriate.
  - In Equation 9, the beta process samples should be in bold face.
  - At the end of line 203/204, it should be RHS instead of LHS.

**Details Of Ethics Concerns:**

Submission3862 has a suspicious resemblance with Submission3881. Both submissions present a variational inference algorithm for computing message aggregation weights in GNNs, modelling the contribution from different neighborhoods as a stochastic process.

1. From the abstracts:
- Submission3862 - *We develop a novel variational inference algorithm to jointly approximate the posterior of the count of neighborhood hops and learn GNN weights while accounting for edge importance.*
- Submission3881 - *We thus propose to model the GNNs’ message-passing behavior on a graph as a stochastic process by treating the number of hops as a beta process. This Bayesian framework allows us to infer the most plausible neighborhood scope for messsage aggregation simultaneously with the optimization of GNN parameters.*

2. Figure 1 in both manuscripts, describing the aggregation strategy, are clearly the same. Moreover, their captions have overlaps:
- Submission3862 - *The sticks located at the top represent random draws from the beta process, serving as layer-wise activation probabilities... The bottom shows the conjugate Bernoulli process.*
- Submission3881 - *The sticks on top are random draws from a beta process, representing the probabilities over the number of hops. The bottom shows the conjugate Bernoulli process over node feature dimensions.*

3. The priors for the models (Equations 3 and 4 in Submission3862, and Equation 2 and 3 in Submission3881) are the same. So are the choices of the variational distributions (Equation 6 in Submission3862 and Equation 7 in Submission3881).

I can point out other similarities, but it is quite clear just by scrolling through the manuscripts that they are different versions of the same work, with Submission3881 being the newer version (eg. with Section 4 on over-smoothing analysis added).

PS. Gave a low score primarily because of the possible dual submission.

---

> ### Author Response · Authors · 2024-11-22
> **Reviewer's Dual Submission Concern**
>
> We thank Reviewer g91e for the thorough and constructive review.
>
> Although both papers utilize Beta and Bernoulli processes for the inference in GNN models, there are significant differences between the two works.
>
> **Methodological Difference:** Our method proposes to apply the stochastic processes directly to the graph for identifying important edges, whereas Paper 3881 applies the stochastic processes to regularize the GNN structures. This is elaborated in the equations as follows:
>
> - Our method: $\mathbf{H}\_{l} = \sigma \left((\widehat{\mathbf{A}}\odot \mathbf{Z}\_{l})\mathbf{H}\_{l - 1}\mathbf{W}\_{l} \right)  + \mathbf{H}\_{l - 1}$
>
>
> - Paper 3881: $\mathbf{H}\_{l} = \sigma(\hat{\mathbf{A}} \mathbf{H}\_{l-1} \mathbf{W}\_{l}) \odot \mathbf{z}\_{l} + \mathbf{H}\_{l-1}$
>
> Note that our method applies the Bernoulli mask samples $\mathbf{Z}_{l}$ on the adjacency matrix. In contrast, Paper 3881 uses the mask on GNN feature activations. This leads to distinct working mechanisms for the two method. Specifically, Paper 3881 prunes neurons in each GNN layer, whereas our method samples graph edges.
>
> Besides edge sampling, we propose to capture the importance of edges using kernel function that updates the drop-edge probability based on the similarity between the node features. However, Paper 3881 assigns equal activation probabilities to all the neurons in the same layer.
>
> **Difference in the Experimental Design:** Regarding the experiments, we focus on examining the effectiveness of kernels for important edge identification and how the subgraphs provide robustness to over-smoothing and over-fitting, whereas Paper 3881 focuses on verifying the theoretical claims it posits.
>
> **Similar figures and phrases:** Although the demonstration figures look similar, they are based on fundamental different methodological ideas. For example, Figure 3 is visualized using the same plotting tool, but it demonstrates different results on different datasets. We will edit the similar phrases in the camera-ready version, as suggested by the reviewer.

---

> ### Author Response · Authors · 2024-11-22
> **Response to the Weaknesses and the Questions**
>
> **"Weak Literature Review":** We mainly discussed the common limitations of the related works in Section 1 Introduction. With respect to our proposed method, these methods are not capable of inferring the number of neighboring hops automatically and require expensive hyper-parameter tuning. The non-Bayesian GNN methods cannot quantify uncertainty in their predictions. Most of the methods in Section 2.3 randomly drop graph edges, which leads to unstable performance. We will clarify the above points and discuss them in both related works and experiments section.
>
> **"No Comparison with Bayesian Methods":** We actually compared our method against the most relevant Bayesian GNN method, BBGDC in Figures 4 and 5 and Table 6. We discussed the method in related work section and called the method "DropConnect (Hasanzadeh, et. al, 2020). In our experiment section, we referred the method with different names, such as "Graph DropConnect" in Figure 4 and "BBGCN" and "BBGDC" in Figure 5. To highlight the advantages of Bayesian methods, such as better calibration and reliable uncertainty quantification, we further benchmarked our method against BBGDC in Table 6, focusing specifically on uncertainty quantification. We will clarify this and use a consistent name in the camera-ready version.
>
> **"The Choice of Beta Process Prior":** Beta process is a complete random measure over integers, allowing the activation probability of the edges in each neighboring hop to go from 0 to 1. This flexible property makes it a better choice compared to other stochastic processes such as Dirichlet process who constraints that all activation probabilities sum to 1.
>
> The monotonically decreasing $\pi_l$  across neighboring hops is due to the stick-breaking construction of the beta process. It reduces the activation probabilities of the edges in longer-range neighborhood scope. However, we can mitigate the effects by setting mild hyper-parameters for the beta process, encouraging smooth decrease. As long as the graph data provide strong support for the edge activations through the GNN likelihood, the settings of beta process wouldn't affect the prediction in any significant way.
>
>
> **Mechanism of $\pi_l$:** $\pi_l$ is a variable about the activation probability of the edges that are exactly $l$-hops away, not up to $l$-hops. It is not defined on node features. This is why the dimensionality of $Z_l$ is consistent, corresponding to the number of edges in the graph.
>
> **"Advantage of using a conjugate Bernoulli-process":** We use variational inference to approximate the marginal likelihood is because of the nonlinearity of the GNN weights and the number of hops going to infinity. We think it's reasonable to extend our Bayesian inference to DropMessage-style message-passing by sampling Bernoulli variables to mask the message matrix.
>
> **"The change in Figure 4 w/o residual connection":** The results of our ablation study in Table 4 shows the performance of our method without residual connections.
>
> **"The model setting of ours in Table 2":** Our method is equipped with the RBF kernel and the architecture is described in Eqn. 1.
>
> We will clarify the above points in the camera-ready version. We will also add the dimension for the matrices and vectors, bold the degree matrix notatio, and reflect the changes made for the mentioned issues in the camera-ready version.
>
> **"Empirical Evidence for Neighborhood Scope Determination":** We have added an analysis on neighborhood scope inference in [this figure](https://anonymous.4open.science/r/iclr2025-D60E/neighborhood%20modeling.pdf) and also in Appendix B.3 in the updated version. We use two synthetic datasets: BA-Shapes and Tree-Cycles in GNN explanation studies [1], and analyze their ground-truth explanations in the form of motifs. The linked figure shows that inferred neighborhood scope highlights the optimal range for extracting meaningful information during training, with second-order neighborhoods being sufficient for BA-Shapes and third-order neighborhoods required for the ring-like motifs in Tree-Cycles.
>
> [1]​​ Ying, Z., Bourgeois, D., You, J., Zitnik, M., & Leskovec, J. (2019). Gnnexplainer: Generating explanations for graph neural networks. Advances in neural information processing systems, 32.

---

> ### Comment · Reviewer_g91e · 2024-11-29
>
> >Note that our method applies the Bernoulli mask samples on the adjacency matrix. In contrast, Paper 3881 uses the mask on GNN feature activations.
>
> This does not make the two works significantly different, but only marginally. The model is exactly the same except for this work employing a DropEdge-like mechanism, while the other employing a Dropout-like mechanism. The motivation for the work is exactly the same, the proposed solutions are very similar, and the choice of the prior distribution is exactly the same.
>
> >For example, Figure 3 is visualized using the same plotting tool, but it demonstrates different results on different datasets. We will edit the similar phrases in the camera-ready version, as suggested by the reviewer.
>
> I didn't raise any concerns about Figure 3. I am more interested in the similarities between Figures 1 of the two works, as well as between Figure 2 of this work and Figure 3 of the other work. They are exactly the same, except for slightly different color choices.
>
> >Most of the methods in Section 2.3 randomly drop graph edges, which leads to unstable performance.
>
> I only see DropEdge and DropEdge++ in the current version of the manuscript. Dropout is also discussed but it has nothing to do with dropping along edges. Rather, more relevant works include DropNode, DropAgg, DropGNN, etc. which are missing from the related works section; although, I do agree that their limitations are similar to DropEdge in that they dropping is performed identically. The most relevant work I am aware of is Graph DropConnect, which has a Bayesian interpretation involving learning of dropping probability along each edge $\times$ feature pair. This addresses the problem this work highlights, is more general and flexible than the dropping algorithms mentioned above, and performs competitively in practice.
>
> > The monotonically decreasing $\pi_l$ across neighboring hops is due to the stick-breaking construction of the beta process.
>
> I understand. I meant that this intuition/motivation should be highlighted in the manuscript so that a reader unfamiliar with the beta process model can infer its relevance to learning via message-passing.
>
> > $\pi_l$ is a variable about the activation probability of the edges that are exactly $l$-hops away, not up to $l$-hops.
>
> I fail to see that since in Section 3.1, there is an addition of self-loops before edge-weights are computed. Wouldn't that imply that masking the edges in the $l$-th step masks information from nodes reachable in $l$-hops, which includes all nodes up to $l$-hops away?
>
> > It is not defined on node features. This is why the dimensionality of $\mathbf{Z}_l$ is consistent, corresponding to the number of edges in the graph.
>
> I apologize, but I don't see what concern is being addressed here.
>
> > We think it's reasonable to extend our Bayesian inference to DropMessage-style message-passing by sampling Bernoulli variables to mask the message matrix.
>
> My concern is that the manuscript says, "The beta process induces hopwise activation probabilities and its conjugate Bernoulli process enables us to adaptively sample the edges in the neighborhood." This suggests that there is an inherent advantage in using the conjugate prior, when there doesn't seem to be one. Also, if not implemented in this work, I would appreciate a remark about the possible extension of this method to DropMessage-like dropping.
>
> > The results of our ablation study in Table 4 shows the performance of our method without residual connections.
>
> Fair enough, that is my bad. My apologies.
>
> > We use two synthetic datasets: BA-Shapes and Tree-Cycles in GNN explanation studies
>
> My apologies, but I am not completely sure why the inferred scope for BA-trees and for Tree-cycles should be 2 and 3, respectively. I understand that the model predicts the scopes to be 2 and 3, but since I am not sure what should be the correct scope, I can't conclude whether the model is correct or not.
>
> Note: Updated my rating, but I think the writing is still not sound. **Importantly, the ethical concern remains**.

---

> ### Author Response · Authors · 2024-11-29
>
> We thank Reviewer g91e for raising the score and providing additional valuable comments.
>
> >**"This does not make the two works significantly different, but only marginally. The model is exactly the same except for this work employing a DropEdge-like mechanism, while the other employing a Dropout-like mechanism."**
>
> We thank the reviewer pointed out the different mechanisms of the two proposed approaches to address the similar problem. Meanwhile, to the best of our knowledge, we are the first propose the notion of adaptively sampling edges and integrate kernel methods into the Bayesian GNNs to evaluate edge importance.
>
> >**"I am more interested in the similarities between Figures 1 of the two works, as well as between Figure 2 of this work and Figure 3 of the other work. They are exactly the same, except for slightly different color choices."**
>
> The two papers make use of a similar visualization tool for the beta processes figure 1 and figure 2 in our paper with that of figure 1 and figure 3 of 3881. However, considering figure 1(a), we show that our model learns edge importance by using different intensity of color within the neighborhood, which is not present in 3881’s figure. Figure 2 in our paper is generated on different dataset from different methods.
>
> >**" The most relevant work I am aware of is Graph DropConnect, which has a Bayesian interpretation involving learning of dropping probability along each edge feature pair. This addresses the problem this work highlights, is more general and flexible than the dropping algorithms mentioned above, and performs competitively in practice."**
>
> We actually compared our method against Graph DropConnet in Figures 4 and 5, and Table 6. We discussed the method in related work section and called the method "DropConnect (Hasanzadeh, et. al, 2020). In our experiment section, we referred the method with different names, such as "Graph DropConnect" in Figure 4 and "BBGCN" and "BBGDC" in Figure 5. To highlight the advantages of the Bayesian GNNs, such as better calibration and reliable uncertainty quantification, we further benchmarked our method against Graph DropConnect in Table 6, focusing specifically on uncertainty quantification. We will clarify this and use a consistent name for DropConnet in the camera-ready version.
>
>
> >**"there is an addition of self-loops before edge-weights are computed. Wouldn't that imply that masking the edges in the -th step masks information from nodes reachable in $l$-hops, which includes all nodes up to $l$-hops away?"**
>
> The reviewer is correct that the addition of self-loops before computing edge weights implies that masking the edges in the $l$-th step affects information from nodes reachable in up to $l$-hops, not exactly $l$-hops. This means that $\pi\_l$​ should be interpreted as the activation probability of edges up to $l$-hops away.
>
> >**Also, if not implemented in this work, I would appreciate a remark about the possible extension of this method to DropMessage-like dropping.**
>
> DropMessage is a quite interesting method. A possible way to extend our method to it is through a hierarchical beta process prior. Specifically, we define a hyper-prior beta process at the global level to adaptively sample a sub-graph when spanning a neighborhood (i.e., corresponding to the size of the Message Matrix) and a local beta-Bernoulli process prior locally on each sampled edge within the neighborhood to regularize the edge-specific message's feature dimension. We will cite and discuss the DropMessage paper and clarify the possible extension as a part of our future work in the camera-ready version.
>
> >**"I am not completely sure why the inferred scope for BA-trees and for Tree-cycles should be 2 and 3, respectively. I understand that the model predicts the scopes to be 2 and 3, but since I am not sure what should be the correct scope, I can't conclude whether the model is correct or not."**
>
> As shown in [this link](https://anonymous.4open.science/r/iclr2025-D60E/neighborhood%20modeling.pdf), the motif of BA-shapes is a sub-graph, where each node requires 2-hop neighborhood to aggregate features from every other node. And, for tree-cycles, the motif is a ring structure, where every node requires at least 3-hop neighborhood scope to accumulate features from the remaining nodes. These motifs can be gold-standard as the optimal neighborhood scopes for these synthetic graphs. We will clarify this in our camera-ready version.
>
> We will add the clarifications as suggested by the reviewer in the camera-ready version.

---

> ### Comment · Reviewer_g91e · 2024-12-03
>
> >We actually compared our method against Graph DropConnet in Figures 4 and 5, and Table 6.
>
> Oh sorry, that's my bad for not noticing! That concern is resolved.
>
> >As shown in this link, the motif of BA-shapes is a sub-graph, where each node requires 2-hop neighborhood to aggregate features from every other node. And, for tree-cycles, the motif is a ring structure, where every node requires at least 3-hop neighborhood scope to accumulate features from the remaining nodes.
>
> Mm okay, I am not confident, but I guess that's not a concern anymore.
>
> **I still think that the ethical concerns remain.** The rest of my concerns are resolved, and I'll update my rating to a 6.

---

> > ### Author Response · Authors · 2024-12-04
> >
> > We sincerely thank Reviewer g91e for raising the score and engaging us in a constructive and insightful discussion. We will make sure all of our discussions will be reflected in the camera-ready version.

---

### Official Review · Reviewer_Sqnu · 2024-11-10

**Soundness:** 2
**Presentation:** 3
**Contribution:** 2
**Rating:** 5
**Confidence:** 4

**Summary:**

In this paper, the authors propose a probabilistic GNN model to address the challenge of determining the optimal neighborhood scope for information aggregation, which is crucial for enhancing GNN performance. The model utilises a beta process to represent neighborhood expansion as a stochastic process, enabling dynamic adaptation of the neighborhood scope. A variational inference mechanism approximates the posterior distribution over neighborhood hops, balancing the neighborhood scope and edge activation for a given dataset. The model adaptively samples edges, identifies significant pathways, and uses feature similarity to evaluate edge importance.

**Strengths:**

- The proposed model introduces a probabilistic approach to neighborhood expansion, which allows for flexible adaptation.
- The use of variational inference to approximate posterior distributions over neighborhood hops is novel.
- The model adaptively samples edges and identifies important pathways, which could potentially improve the quality of information aggregation.

**Weaknesses:**

Overall, my primary concern with this paper is the insufficiency and lack of convincing experimental validation. The primary experimental validation is conducted on small-scale datasets (e.g., Cora, Citeseer, Pubmed), which are known to be too limited for drawing strong conclusions in GNN research. These datasets are also prone to high variability due to different model initialisations, making the results less convincing. More experiments on large-scale datasets, such as those in "OGB" or "benchmarking graph neural networks", are needed to support the claims.

Also, the authors refer to ogbn-arxiv and ogbn-mag as "large-scale" datasets, but these are officially classified as "small-scale" and "medium-scale" in the OGB benchmark. Furthermore, these datasets are only used in a limited portion of the experiments (Tab. 3).

While the authors discuss over-smoothing measurements, Dirichlet energy is, in fact, more widely adopted in the literature compared to the total variation (TV).

In terms of methodologies, the paper misses the discussion of related work on path-based aggregation in GNNs, which addresses similar challenges (e.g., "Path Neural Networks: Expressive and Accurate Graph Neural Networks", ICML 2023).

Furthermore, the use of feature similarity to assess edge importance is straightforward, and it is unclear whether this approach is effective for larger datasets. This raises doubts about whether feature similarity alone can indeed accurately represent edge importance. It would be great if the authors could provide more insights on this point.

(Minor) Line 142: The sentence "Since GNN layer l aggregate information within l-th neighborhood" contains a grammatical issue.

**Questions:**

1. Could the authors provide more insights into the limitations of using feature similarity as the sole metric for edge importance, especially for large-scale datasets?
2. How does the proposed method compare with the related path-based aggregation methods, such as those described in the ICML 2023 paper mentioned?

---

> ### Author Response · Authors · 2024-11-25
>
> We appreciate the reviewer’s constructive comments on our work which are valuable for our future research.
>
> **"More experiments on Larger-scale Datasets":** To address the reviewer's concern, we conduct additional experiments on OGB datasets, including ogb-proteins and ogb-products. Although they are listed as medium-scale ones, they have a larger number of edges and nodes in comparison with OGB-mag we presented in the submission. We reported a portion of these experiments on these larger datasets in the tables below.
>
> Table 3: Test accuracy (%) comparisons with larger datasets on semi-supervised node classification task
>
> | Dataset          | GCN  | ResGCN  | JKNet | GCNII | **GAT**       | **Ours**                 |
> |------------------|------|---------|-------|-------|---------------|--------------------------|
> | ***ogb-Proteins***    | *72.5* | *73.4* | *69.5* | *74.1* | **85.0** | *83.6 ± 0.3* |
> | ***ogb-Products***   | *82.3* | *82.5* | *82.9* | *83.7* | *81.7* | *83.8 ± 0.4* |
>
> The experiments highlight our method’s scalability when it comes to larger-scale datasets. We have also included the results of these experiments in the updated version of the manuscript.
>
>
> **Extensive Experiments on Larger-Scale Datasets:** We have revised our manuscript to reflect the scale of datasets as pointed out by the reviewer. We have further extended our experiments to incorporate the comparison on the medium scale datasets. The added results are summarized in the tables below. Additionally, the updated results in tables 3, 4, 5 and 6 reflect the extensions made in the paper's updated version.
>
> Table 4: Ablation study of different modules' effectiveness in our model
> | Module                                  | Cora                | Citeseer            | Pubmed              | *ogb-Arxiv*          | *ogb-Mag*           |
> |-----------------------------------------|---------------------|---------------------|---------------------|---------------------|---------------------|
> | **Ours**                                | **83.2 ± 0.5**      | 71.5 ± 0.3         | **78.5 ± 0.2**      | ***75.2 ± 0.4***    | ***44.3 ± 1.7***    |
> | w/o kernel                              | 82.2 ± 1.2         | **71.7 ± 1.1**     | 77.9 ± 0.6         | *74.3 ± 1.1*        | *42.7 ± 3.1*        |
> | w/o beta process                        | 79.4 ± 0.3         | 67.8 ± 0.2         | 77.9 ± 0.2         | *72.3 ± 0.2*        | *43.1 ± 0.7*        |
> | w/o skip-connection                     | 81.2 ± 0.4         | 69.8 ± 0.1         | 77.6 ± 0.3         | *71.5 ± 0.6*        | *42.9 ± 0.4*        |
> | w/o kernel, beta process, and skip-connection | 78.7 ± 0.2         | 66.2 ± 0.3         | 77.5 ± 0.2         | *70.2 ± 1.3*        | *37.2 ± 0.6*        |
>
> Table 5: Semi-supervised node classification training time comparison
>
> | Methods        | Time (Cora) | Space (Cora) | Time (Citeseer) | Space (Citeseer) | Time (Pubmed) | Space (Pubmed) | Time (*ogb-Arxiv*) | Space (*ogb-Arxiv*) | Time (*ogb-Mag*) | Space (*ogb-Mag*) |
> |----------------|-------------|--------------|-----------------|------------------|---------------|----------------|--------------------|---------------------|------------------|-------------------|
> | **GCN**        | 62.53       | 39           | 77.37          | 164              | 96.54         | 129            | 643.72            | 2421               | *2765.45*        | *4805*           |
> | **GCNII**      | 60.14       | 42           | 79.22          | 175              | 95.76         | 133            | 614.37            | 2525               | *2840.32*        | *4953*           |
> | *JKNet*        | *61.27*     | *41*         | *80.15*        | *184*            | *100.33*      | *136*          | *661.44*          | *2606*             | *2911.59*        | *4904*           |
> | **Ours**       | 67.38       | 57           | 88.51          | 194              | 102.36        | 166            | 677.58            | 2788               | *3033.75*        | *5277*           |
>
> Table 6: Evaluating the uncertainty estimation of models with ECE metric
>
> | Model   | *ogb-Arxiv*           | *ogb-Mag*            |
> |---------|-----------------------|----------------------|
> | **GCN** | *8.43 ± 1.22*         | *6.56 ± 0.97*        |
> | **GCNII** | *10.04 ± 0.08*      | *6.92 ± 0.04*        |
> | **Ours** | ***5.02 ± 1.03***    | ***3.53 ± 0.88***    |

---

> ### Author Response · Authors · 2024-11-26
>
> **Dirichlet Energy Experiment**: We report the experimental results using Dirichlet Energy as a metric for over-smoothing analysis in [this link](https://anonymous.4open.science/r/iclr2025-D60E/dirichlet%20energy.pdf) and also added the result in Figure 7 in the updated version. We achieve larger Dirichlet energy between the node features across the truncation level compared to the baseline methods. This result is consistent with what we reported using total variation.
>
> **Comparison with PathNN-based GNN**: We will add the discussion about PathNN-based GNNs [1] in the related work in the camera-ready version. PathNN-based GNN focuses on graph classification, whereas our experiments primarily target node classification. Extending our work for graph classification will be a part of our future research.
>
> **Effectiveness of Edge Importance on Larger Datasets:** We evaluate the effectiveness of our method on larger datasets: **ogb-Arxiv** and **ogb-Mag** through an ablation study presented in table 4, which are in the table below. The results demonstrate that without edge importance the method performs significantly worse compared to with edge importance. This highlights the effectiveness of inferring edge importance on larger datasets.
>
> | Module                                | **ogb-Arxiv**          | **ogb-Mag**           |
> |---------------------------------------|---------------------|---------------------|
> | **Ours**                              | **75.2 ± 0.4**   | **44.3 ± 1.7**    |
> | Ours w/o kernel                            | 74.3 ± 1.1        | 42.7 ± 3.1        |
>
> We have fixed grammatical issues in line 142 in the revised version. We will include the changes in the camera-ready version.
>
> [1] Michel, G., Nikolentzos, G., Lutzeyer, J., & Vazirgiannis, M. (2023). Path neural networks: expressive and accurate graph neural networks. In Proceedings of the 40th International Conference on Machine Learning.

---

> ### Author Response · Authors · 2024-12-02
> **Gentle Remainder**
>
> We thank the reviewer for the constructive feedback and would like to respectfully remind the reviewer that the deadline for the discussion period is approaching. if the reviewer has any other questions, we would be glad to address them. Thank you again for your time and thoughtful review!

---

> ### Author Response · Authors · 2024-12-03
>
> As the discussion period is nearing its conclusion, we sincerely thank the reviewer for the valuable feedback. We have made every effort to address all concerns and questions raised. If the reviewer feels our revisions meet the expectations, we kindly request consideration for an updated rating. If there are any further questions or matters to discuss, please don’t hesitate to let us know.

---

### Meta-Review · Area_Chair_EkwN · 2024-12-18

**Metareview:**

This paper introduces a novel probabilistic model for Graph Neural Networks (GNNs) that adaptively samples edges to enhance neighborhood information aggregation. The authors aim to reduce computational overhead while improving the performance of GNNs, particularly in node classification tasks.

Reviewers expressed mixed opinions, with one acknowledging the uniqueness of the proposed methodology but still rating it marginally below the acceptance threshold. However, the majority shared apprehensions about the insufficient experimental validation and lack of suitable baseline comparisons. Most reviewers noted that the literature review was weak, failing to connect the proposed methodology with existing GNN techniques effectively.

Following the authors' rebuttal, reviewers' concerns remained largely unaddressed, particularly regarding the effectiveness of the model on heterophilic datasets and the inadequacy of the experimental framework. The persistent criticism centered around the paper's comparative analysis, which was deemed insufficient, particularly in light of other state-of-the-art methods.
Given the overwhelmingly negative feedback from all reviewers and the authors' inability to satisfactorily resolve the outstanding issues, I recommend rejecting the paper. While the authors attempted to mitigate concerns presented during the review process, the substantial limitations identified by reviewers were not sufficiently alleviated.

**Additional Comments On Reviewer Discussion:**

Reviewers expressed mixed opinions, with one acknowledging the uniqueness of the proposed methodology but still rating it marginally below the acceptance threshold. However, the majority shared apprehensions about the insufficient experimental validation and lack of suitable baseline comparisons. Most reviewers noted that the literature review was weak, failing to connect the proposed methodology with existing GNN techniques effectively.

Following the authors' rebuttal, reviewers' concerns remained largely unaddressed, particularly regarding the effectiveness of the model on heterophilic datasets and the inadequacy of the experimental framework. The persistent criticism centered around the paper's comparative analysis, which was deemed insufficient, particularly in light of other state-of-the-art methods.
Given the overwhelmingly negative feedback from all reviewers and the authors' inability to satisfactorily resolve the outstanding issues, I recommend rejecting the paper. While the authors attempted to mitigate concerns presented during the review process, the substantial limitations identified by reviewers were not sufficiently alleviated.

---

### Decision · Program_Chairs · 2025-01-22

Reject